# Creating a Design Framework to Diagnose and Enhance Grassland Health under Pastoral Livestock Production Systems

**DOI:** 10.3390/ani12233306

**Published:** 2022-11-26

**Authors:** Fabiellen C. Pereira, Carol M. S. Smith, Stuart M. Charters, Pablo Gregorini

**Affiliations:** 1Department of Agricultural Science, Faculty of Agriculture and Life Sciences, Lincoln University, Lincoln 7674, New Zealand; 2Centre of Excellence Designing Future Productive Landscapes, Lincoln University, Lincoln 7674, New Zealand; 3Department of Soil & Physical Sciences, Faculty of Agriculture and Life Sciences, Lincoln University, Lincoln 7674, New Zealand; 4School of Landscape Architecture, Faculty of Environment, Society and Design, Lincoln University, Lincoln 7674, New Zealand

**Keywords:** design, grassland, grazing management, health, pastoral livestock production systems, sustainability

## Abstract

**Simple Summary:**

Human life depends on the provision of ecosystem services from grasslands ecosystem, which have been threatened due to practices adopted by current farming systems. Researchers have thus pointed out the use of systems thinking and design theory to create more sustainable pastoral live-stock production systems that work within grasslands for the continuous delivery of multiple ecosystem services. Systems thinking is a holistic theory that brings different perspectives from a particular situation and design theory is a prescriptive theory that aims to achieve a defined goal by using specific tools and background. In this narrative review, we explain why the use of both theories would contribute to the creation of more sustainable livestock production systems and provide an overview of grasslands with the goal of creating the required design framework for the design of pastoral livestock production systems that enhance grassland health.

**Abstract:**

Grasslands and ecosystem services are under threat due to common practices adopted by modern livestock farming systems. Design theory has been an alternative to promote changes and develop more sustainable strategies that allow pastoral livestock production systems to evolve continually within grasslands by enhancing their health and enabling the continuous delivery of multiple ecosystem services. To create a design framework to design alternative and more sustainable pastoral livestock production systems, a better comprehension of grassland complexity and dynamism for a diagnostic assessment of its health is needed, from which the systems thinking theory could be an important approach. By using systems thinking theory, the key components of grasslands—soil, plant, ruminant—can be reviewed and better understood from a holistic perspective. The description of soil, plant and ruminant individually is already complex itself, so understanding these components, their interactions, their response to grazing management and herbivory and how they contribute to grassland health under different climatic and topographic conditions is paramount to designing more sustainable pastoral livestock production systems. Therefore, by taking a systems thinking approach, we aim to review the literature to better understand the role of soil, plant, and ruminant on grassland health to build a design framework to diagnose and enhance grassland health under pastoral livestock production systems.

## 1. Introduction

The central concept of health is the ability of a system and its parts to self-organise its key functions and maintain its integrity in the presence of stress [1,2]. Accordingly, the definition of grassland health should consider its individual components, their interaction, and their connection with the socio-ecological context in which they are embedded. Grasslands are heterogeneous and complex ecosystems, either natural or influenced by humans and driven by biotic and abiotic factors and processes. The biotic processes are derived predominantly through solar energy converted into chemical compounds by photosynthesising plants [3], that depend on abiotic factors (water, temperature, nutrients, and location, as determined by climate soil, and topography), thus determining essential ecosystem services that sustain grasslands [4]. Ecosystem services are all the ecosystem functions and products that benefit and sustain human survival, life, and well-being, including the provision of food and water, pollination, regulation of climate, water and air, primary production and nutrient cycling, habitat for organisms, genetic resources, pharmaceuticals, and aesthetic/cultural values [5,6,7]. Therefore, grassland health can be defined as the ability of soils, microorganisms, flora, and fauna (including humans and grazing animals) present in the system to evolve, adapt and maintain their integrity in the presence of natural or human-caused stress/disturbance and provide essential components to sustain ecosystems [8]. 

Human life is highly dependent on grasslands due to the provision of ecosystem services, but this dependency relationship between man and land has been putting grassland health under threat. As a means to facilitate access of humans to the food and natural resources that grasslands offer, grasslands have been converted to agricultural areas of cropland and pasture [9], resulting in the existence of farming systems, such as pastoral livestock production systems. However, this conversion and common practices adopted by modern livestock farming systems (e.g., frequency and intensity of grazing, excessive fertilizer use, etc.) have been degrading ecological conditions across the globe, thus causing extensive environmental damage [10,11]. For example, domestic ruminant livestock is the source of 48% of the total greenhouse gases emitted from the agriculture sector globally, due to enteric methane (CH_4_ [12]). Around 28–75 billion tons of soil on the planet is lost annually through erosion, partly due to human actions [13]. A loss rate of 20–30% of vegetation species richness has been reported across different biomes around the world [14], thus causing important pollinator populations such as bees to decline [15]. In addition to these examples, other events such as deforestation of tropical areas [16,17] and intensive use of non-renewable natural resources, such as fossil fuels for energy and water [18]. Pollution of soil and water is also of major concern [10,11,19].

Current pastoral livestock production systems need to be carefully transformed to evolve continually within grasslands and improve the sustainability, stability, and resilience of these ecosystems in the future. This transformation demands the understanding and use of sustainable land management strategies that deal with environmental challenges that threaten grassland health [20]. To promote management change and innovation toward the development of more sustainable and context-appropriate systems, the concept of design theory has been largely applied as a new paradigm for farming systems [21,22]. Design theory encompasses principles and predictable procedures based on scientific knowledge for devising artefacts with specific proprieties that meet outlined goals [23]. Design is a participative process [24] that involves thinking, science, local knowledge [25] and interdisciplinarity, adjusting farming systems to a social-ecological-geographic perspective [26]. Sustainable environmental management strategies must be applied to pastoral livestock production systems to efficiently meet growing demands for livestock products while enabling grasslands to continually deliver multiple ecosystem services [19,27]. Therefore, design theory could be used to create alternative and more sustainable pastoral livestock production systems to preserve and/or enhance grassland health. 

To apply the design theory concept to achieve alternative and more sustainable pastoral livestock production systems, we borrowed the theory of design according to Walls et al. (1992 [28]). Design theory is a prescriptive theory based on knowledge towards generating means, methods, and products to attain specific goals and face pre-defined problems. For example, pasture swards can be designed to provide ruminants a biochemically rich forage to enhance individuals and environmental performance [29,30]. Likewise, ruminants can be designed to produce healthy and high-quality products with less environmental impact and better nutrient use efficiency [31]. The design theory described by Walls et al. (1992 [28]) encompasses two features (Figure 1). The first one deals with the product, per se. It involves the construction of a framework depicting the meta-requirements (goals) that addresses a class of problems, the meta-design that describes the artefacts hypothesised to meet the meta-requirements, and the testable design process hypothesis that verify whether the meta-design satisfies the meta-requirement. The second feature deals with the design process; it addresses the design method that describes the construction of the artefacts, the kernel (fundamental) theories that govern the design process itself and the testable design process hypothesis that verifies whether the design method and the meta-design are consistent. 

To apply the design framework proposed by Walls et al. (1992 [28]) to design alternative and more sustainable pastoral livestock production systems to preserve and/or enhance grassland health, a better understanding of grassland health definition, diagnosis, and assessment is needed. This suggests the comprehension of how interactive processes grassland components–soil, flora, and fauna—under climate and topography factors and in response to grazing management and herbivory perform to offer ecosystem services, which is the main objective of this manuscript. In this context, we start by pointing out that systems thinking theory would contribute to understanding the complexity and dynamism of grasslands. Then, by using the systems thinking approach we reviewed the literature to describe each of the key components of grasslands under pastoral livestock production systems (soil, plants, ruminants) in separate sections to outline and determine the meta-requirements, the meta-design, and the kernel theories required for constructing a design framework to achieve the design goal of enhancing grassland health under pastoral livestock production systems.

### Systems Thinking Theory

Current farming systems models result from the necessity of expansion and intensification to supply the increased demand for agriculture products due to the rapid growth and transformation of human societies since the 18th century [9,10]. As the focus was to increase productivity and have quick and straightforward outcomes, a reductionist approach was adopted, which was successful and well suited to the context of increased agricultural yields [10,11,32]. The reductionist approach implies a linear view, which inappropriately perceives grassland systems as being static and equilibrium-centred [32]. However, grasslands are complex and dynamic ecosystems that change in space and time, whose effects of change are not linear [33,34,35]. Such dynamism may lead to uncertain outcomes and unpredictable collateral effects elsewhere from any external/human-related actions taken [36]. Consequently, the improper use of grasslands has led to ecosystems degradation, to which livestock production systems are a major contributor [19]. The negative results of applying a reductionist approach to farming systems suggest that the design of alternative and more sustainable farming systems demands a shift from a solely reductionist approach to one that accounts for the complexity and dynamism of grasslands.

A more holistic approach would contribute to a clearer understanding of how grassland components and processes interact toward grassland performance and enhanced health. Systems thinking, as a theory, uses a holistic view that aims at integrating approaches and understanding the non-linear relationships among the components and processes of a system, as a means of dealing with its complexity and disequilibrium in different dimensions [37]. As ‘problematic’ situations that affect grasslands cannot be addressed by using a reductionist approach only, systems thinking theory is imperative for designing more sustainable and alternative pastoral livestock production systems [38]. In pastoral livestock production, grazing management drives and sustain the system productivity, and so it should be planned based on an advanced understanding of grassland component interactions and ecological processes to reveal the multiple ecosystem services provided by grasslands. Systems thinking would contribute to uncovering multiple perspectives towards grassland systems, their components, and their connection with ecosystem natural processes. It provides an overview of a system and its behaviour to facilitate the comprehension of how grassland components drive ecological processes to assess and enhance grassland health. 

Systems thinking is a context-focused theory in which the relationship between the system properties and its effects on what surrounds it explains how the system works [39]. This corresponds perfectly with grasslands, as grassland health, although being a general concept, is highly dependent on the context. For example, a particular classification of chemical and physical properties of soil can be considered adequate for plant growth in some areas while totally inadequate for others; some plants considered to have low production may grow better in a place where climatic conditions are more appropriate to those species, thus influencing the system primary productivity, and other factors such as the system ability in sustaining livestock production. Systems thinking is considered a diagnostic approach to dealing with complex situations [40], where the aim is the resolution/improvement of a ‘problem’ according to each situation, rather than a single and generalized solution [39]. Accordingly, there is not only one strategy to preserve/enhance grassland health, but more than one, and these are divergent to different situations. A particular form of same grazing management may result in increased quality of pasture in one place [41] but result in negative outcomes elsewhere [42]. As it is about connection, interaction, relationships, and co-evolution, systems thinking builds on interdisciplinarity for a better system performance rather than looking into individual parts of the system, such as only crop yield [38]. 

Systems thinking uses rich pictures as illustrative tools to explore and express situations, which would aid in the construction of the design framework of the systems thinking approach. Rich pictures are preliminary diagram representations that require participatory work from stakeholders with different points of view used to clarify the complexity and messiness of systems by facilitating the visualization of ‘problematic’ situations [43,44]. Through communication, participants depict their perceptions of all the interconnections, subjective elements, and non-linear structures within a system [44]. This process also involves co-learning and deeper insights from the situation, which enhances the understanding and/or uncovers other gaps that need to be analysed [45]. A rich picture exposes a system to be seen as a whole, thus facilitating research and sensemaking [45], besides implementing an interdisciplinary approach. Readers are invited to see Pereira et al. (2021 [8]) as an exercise of using rich pictures to enable discussions between different researchers with different backgrounds and disciplines to graphically represent grasslands and identify grassland components and processes to diagnose and better understand grassland health. Using systems thinking theory as a base approach, therefore, is paramount for the diagnostic understanding of grassland health for the construction of a framework that will guide the design of alternative, sustainable, and ethical livestock production systems [46]. The following section explores the role of the key components of grasslands (soil, plants, and ruminants) and their interaction on grassland health and how they contribute to the design process. 

## 2. Systems Components 

### 2.1. Soil 

Soil is essential in the most varied ecosystems for providing ecosystem functions and services [47]. Besides supporting primary production through nutrients and water supply to provide food, fiber, and medicine [48], soils also contribute to regulation services such as climate, water, and nutrient regulation [49], C sequestration and storage, flood and pest control, and cultural services in terms of encouraging its sustainable use [50]. Soil influences and supports the community of plants, ruminants, and microorganisms occurring both above and below ground [51]. Through soil, plants supply carbohydrates, root exudates and detritus that support microbial life, such as bacteria and fungi, and benefit from the nutrients released from them [3]. By grazing, ruminants influence the renewal and growth of plants due to defoliation [52] and the return of nutrients to the soil from defecation [53], thus contributing to soil nutrient cycling, C input, ground cover and the soil micro and macro biome [54,55]. This interdependent relationship between soil, plants, and ruminants, besides determining soil suitability for performing ecosystem services and functions, determines the resilience capacity of ecosystems [4,48], and mediates soil health [56].

Soil health is defined as the capacity of soils and their emergent properties to self-organise, perform and function to provide ecosystem services to fulfil plant, ruminant, and human demands [50,56,57]. The assessment of soil health is still not completely clear, but some soil indicators, involving chemical, physical, and biological attributes, and the consequences of those on soil functions are commonly used measurements. This implies that different areas of soil science can be used as Kernel theories for the design framework. Physical parameters of soil, such as the spatial arrangement of aggregates, and porosity are mainly associated with its structure [58] but are determinants of water and solute transport [59], thus influencing other chemical, hydrological, and biological functions of the soil. Chemical parameters including cation exchange capacity, pH, and C content, are related to processes such as nutrient cycling and plant growth, and biological parameters such as microbial biomass and organic matter, thus promoting biological activity and root development [56,57]. Those indicators define a soils ability to function, which is partly dependent upon soil formation but also explained by climate and topographic conditions, and the consequences of the management practices applied to the soil for specific agricultural purposes [57,59]. Therefore, the health of a soil depends upon the interaction between its emergent properties in a particular context. 

Climate and topographic features are two important factors that simultaneously affect soil properties and processes, and therefore, soil health. In areas with varying steepness of slope, chemical parameters are usually naturally more concentrated at the foot slope due to the downward movement of water and minerals from the upper slope position [53]. This can be confirmed for levels of pH, exchangeable calcium, and magnesium in a subtropical rain forest in Taiwan, but not for aluminium, which was higher in the upper slope soils [60]. Physical properties tend to be more susceptible to damage in top areas [53]. Consequently, higher altitude areas are usually more affected by degradation, which explains why the shoulder slope position showed greater improvement in soil health over time in a study of grassland reestablishment in the United States [61]. Similarly, different hill slope aspects imply different microclimate conditions on soil such as moisture and temperature that affects soil nutrients, organic matter, and microbial activity [62]. Aspect also affects the degree of shading, which influences vegetation cover composition. Covered areas have more stable soil aggregates due to the greater amount of organic matter entering the soil, which is beneficial for the root development of plants [63]. Climate factors also affect soil properties. In southern Bolivia, the aggregate stability of soil was higher the higher the mean annual rainfall [64]. However, in arid environments, increased rainfall is associated with high rates of runoff and erosion [65], thus confirming that the assessment of soil health is dependent upon its geoclimatic context. 

Soil health is also a consequence of its historical land use, as it is affected by impacts caused by human management decisions [56,57]. In pastoral livestock production systems, grazing factors such as stocking density and grazing time influence soil properties and soil dynamics in multiple ways. When comparing ungrazed areas with areas grazed by sheep at different intensities (lightly, moderately, or heavily grazed) in Mongolia, bulk density increased in all the grazed areas regardless of the intensity, but no differences were noticed in soil chemical parameters [66]. However, another study performed also in Mongolia [67] with lower stocking densities and different grazing times during the year (e.g., winter grazing only or year-round grazing with higher intensity during summer), indicated not only a difference in bulk density, but lower organic C, total N, and total S concentrations in areas with higher grazing intensities. Results from a literature review comparing grazed and ungrazed sites around the world show that organic C in the soil may increase, decrease, or remain unchanged depending on grazing and climate conditions [68]. Nevertheless, the authors observed some general patterns, such as greater root content (which controls the formation of soil organic C) in grazed areas compared to ungrazed areas at the driest and wettest sites, lower root content at sites with intermediate precipitation, and increased rates of C:N ratios in grazed areas. The contrasting results from these studies, besides reinforcing the dependency of soil properties to a geoclimatic context, indicate that the influence of grazing can occur in complex ways. 

As soil is affected by management, its health is threatened by degradation processes due to human pressures caused by uncontrolled and inappropriate land use [69]. One of the biggest factors from grazing activities influencing soil properties is treading. Treading is the pressure effect of ruminants on the soil, which is a function of the ruminant mass, foot size and kinetic energy [70]. Treading affects mainly soil physical properties, as observed in areas grazed by dairy cows in different soil types of the North Island of New Zealand [71] but consequences may extend to soil hydrological functions, such as water infiltration rates, even at low grazing intensity [72]. Those effects are dependent upon grazing intensity (number of ruminants grazing at the same time in a given area) and variation in topography and spatial heterogeneity of vegetation, which implies different treading pressure over the soil due to the distinctive distribution of grazing over the land. Overgrazed areas affect the regrowth and restoration capability of plants, which decreases soil groundcover and increases soil physical and biological degradation [73], possibly leading to long-term processes such as compaction or pugging [70]. Soil compaction is a reduction in soil pore space, hydraulic conductivity, and/or surface soil infiltration [74] while pugging in the remolding of wet soil around the hoofprint of the ruminant [70] with consequent loss of soil strength [75]. Although different processes, both influence soil physical properties, such as bulk density, porosity, penetrometer resistance, and permeability to air or water [58,75]. Decreased soil porosity and increased bulk density are observed after heavy treading events due to soil compaction, affecting soil air and water transmission and root growth [53,76]. Those parameters influence soil hydrological functions, which are essential for regulating soil water dynamics and plant water-use efficiency [76], with further consequences for soil health and its ability to support plant growth. 

Where the ability of soil to support plant growth is jeopardized, vegetation cover is reduced and soil susceptibility to erosion increases, which is detrimental to soil health. Vegetation cover attenuates the impact of rainfall on soil, increasing soil infiltration and surface cohesion and strength, and controlling the surface flows, thus reducing run-off and sediment loss [77]. Vegetation cover also acts as a buffer, protecting soil against treading pressure and preserving its physical qualities [78]. Where vegetation cover is reduced, soil protection is undermined and its susceptibility to erosion increases [74], thus decreasing organic matter content and impairing soil structure, biota, and cation exchange capacity [79]. A reduction in organic matter inputs with an increased proportion of bare soil affects the soil microclimate and accelerates soil evaporation rates [80] due to fluctuations in soil moisture and temperature [81]. This not only affects soil microbiota activity [82] but promotes the upward movement of water and soluble salts to the topsoil causing salt accumulation [80], and impairing vegetation growth. As soil erosion degrades soil structure and depletes soil nutrients due to decreased soil water storage capacity and decreased organic matter [79], soil productivity and function is decayed and further negatively affects ecosystem services [83]. If prolonged erosion events happen, soil loss may be irreversible and its ability for biomass production and its filtering capacity is depleted [79]. Therefore, the complex relationship between soil and vegetation should be considered in the design process to guarantee the construction of artefacts that assure positive impacts for soil health and the provision of ecosystem functions. 

Plants contribute to the soil regenerative process after deterioration, protect the soil against further disturbances, and positively impact soil health. Vegetation is crucial to prevent soil exposure and avoid erosion and compaction [4], besides playing an important role in improving the dynamics of the biota community in soils, which is essential for several ecosystem functions. Through defoliation, plant root exudation stimulates organic matter decomposition that promotes the rhizosphere microbial population and activity, which liberates limiting nutrient such as N for plant regrowth [84,85], and improves soil structural stability and resilience [76]. The decomposition of organic matter by microbial activity in soils provides greater stability of soil aggregates and stronger resistance to treading damage, thus reducing the susceptibility of soil to compaction and surface runoff [86,87]. Microorganism activities also include the transformation of organic matter into nutrients, which influence the chemical and physical composition of the soil, promoting nutrient cycling and nutrient assimilation by plants [86,88]. The benefits from plant roots in soil are maximised in multispecies swards [89] due to the mutualistic symbionts occurring between the different microbial communities around the roots of different plants [90]. The interdependence relationship between soil, plants, and organisms determines soil suitability for primary production [48], which besides mediating soil health [91], is an important ecosystem service provided by grasslands [4] and essential for sustaining pastoral livestock production systems. 

Soil health is linked with ecosystem resistance and resilience capacity [4,48,92] and is essential to assure the best performance of pastoral-livestock systems due to its contribution to providing ecosystem functions. As a component of grasslands, soil health strongly influences grassland health, so grazing management that assures soil productivity function and health is key for enhanced grassland health and designing more sustainable pastoral livestock systems. Adequate management should include the enhancement of soil processes in the meta-requirements of the design framework, such as decreased soil degradation and enhanced ecosystem functions, including organic matter accumulation, water infiltration, and nutrient cycling [52]. Accordingly, practices that account for the interaction between soil abiotic properties, soil biota, and plants, are paramount for the health status of a soil, for example the use of a biodiversity of swards, minimum soil disturbance, and positive influence from ruminants through defoliation and excretion. As soil features and processes are highly context-dependent, optimal management for soil health needs to consider local geo-biophysical, socio-economic, and climatic conditions [49,59]. Different grazing management reflects variability in grazing activities that influence soil health (defoliation, excretion, and treading), with potentially different effects in different environments. That is the ultimate reason why grazing management needs to be adjusted for the local soil emergent properties and processes that determine the land capability and suitability to support grazing pressure and assure a sustainable pastoral livestock production system. 

### 2.2. Plants

Plants are major components of grasslands, as drivers of biogeochemical cycles and ecological processes that sustain grassland health and promote their ecosystem services. Grasslands are composed of distinct types and communities of grasses, forbs, shrubs, and a small proportion of trees that use solar energy for primary production [7]. Primary production is the energy source to sustain food chains in ecosystems, and thus, essential for any life support. Through this process of solar energy conversion, plants capture carbon from the atmosphere to produce forage that will feed and medicate ruminants, which also contributes to climate regulation, besides mediating soil health by increasing its structural stability, protecting it against erosion, driving nutrient dynamics and availability and promoting soil biocenosis, as previously stated [5,6,7]. Consequently, plants are strongly related to three key ecological attributes of ecological systems: vigour, organization, and resilience, used as measurement indicators to quantitatively assess grassland health [4,93,94]. Vigour is the primary productivity of a system, organization indicates the biodiversity and interaction of components in a system, and resilience is the capacity of a system in maintaining its structure and function over time in the face of external disturbance [95,96]. Those attributes explain and measure the multi ecological processes occurring between plants and other grassland components—soil and ruminants—that sustain grasslands and define grassland health [93,94,96]. Thus, we believe that vigour, organization, and resilience are important concepts to be considered when defining the kernel theories for the design framework. 

The vigour of grasslands is the amount of biomass produced by each of the vegetation types in terms of density, quality, and yield, which quantify the magnitude of the energy available for the ecosystem activity, which in this case, is primary production [4,94]. The large variation in primary production across grasslands is due to a natural genetic variation in plants and their phenotypic interaction with different environments [97]. A plant species individual ability in producing biomass is dependent upon its ecophysiological traits, such as specific leaf area, chlorophyll concentration, leaf dry matter content, leaf nitrogen content [98,99], and leaf photosynthetic nitrogen use efficiency [100]. These traits are a consequence of canopy structure and site features (fertility, climate, management), affecting C fluxes and consequently, indicating a plant capability in intercepting light for a maximum growth during photosynthesis [99,101,102]. The physiological traits vary even between leaves in the same plant [97], and its relationship with the plant photosynthetic capacity varies among and within species and functional groups, are scale-dependent, and change through seasons [98]. Differences in plant traits are also found in plants with different photosynthetic pathways, (i.e., C_3_ or C_4_), thus affecting not only the system production efficiency but also the C storage on soils [103]. Accordingly, the vegetation community and its growth dynamics are highly variable between and within environments, thus affecting grassland ecological processes and services, including primary production [102]. 

Large heterogeneity that has been observed in plant community traits due to species diversity is a consequence of varying environmental features in the various contexts [104,105,106]. A given environment results from a geological formation, which determines soil and topographic features and different elevation gradients [107], consequently influencing other abiotic factors—temperature, humidity, slope, aspects, and geological substrates [106,108] which all influence plant traits. In the Mediterranean region, trees, shrubs, and forbs were found along the whole climate range but their abundance and pattern traits, such as leaf size, texture, thickness, and type, changed according to climatic gradients [109]. For example, there was a greater abundance of small leaves in cold and dry environments, and the more arid the environment, the more coriaceous and thicker the leaf texture. In an alpine meadow environment on the Qinghai-Tibetan Plateau, plant communities with distinct root morphology (fibrous or tap) differed between sunny and shady slopes due to differences in soil moisture and temperature created by greater incident solar radiation on sunny aspects, which consequently resulted in differences in vegetation biomass [62]. Besides geological and abiotic features, landscape appearance is also an adaptative response of flora and fauna to external disturbances over time, such as land use and fire [107]. In the South Island of New Zealand, the change in vegetation due to fire events that occurred prior to human occupation and the recent emergence of pastoralism has caused a shift in the dominance of tussock grasses for other species (indigenous or not) more tolerant of grazing activities [110]. Hence, the combination of all the aforementioned factors defines the heterogeneity and distinction of grasslands around the world. 

Plant species have an individual ability for producing biomass but the production resultant from the complex community dynamics of all the vegetation groups present is what defines grassland productivity. As plant species characteristics such as root allocation and length are involved in nutrient acquisition and availability in different ways (associated with above and belowground biomass), the more diverse the species composition of a system, the more functional it is [111]. Functional diversity increasessystem productivity due to the complementary function among groups, which also influences soil biota and benefits ecosystem services and processes, such as nutrient cycling and C sequestration [112]. Greater diversity is usually desirable due to the enhanced benefits that the dynamic interaction of several plant species with different specific traits provides to the system compared to individual species or different species with similar traits. However, a positive multifunctionality is achieved when species whose traits are functional complementary interact synergistically [113]. For instance, Husse et al. (2017 [114]) reported the effects on nutrient uptake of mixed species differed with plant traits. Positive effects were noticed when combining N_2_ fixing species and non-fixing species with shallow roots, but they were negative for deep-rooting species. Even larger positive effects were noticed when both traits were combined: species that differ in their nitrogen-fixing ability and in their rooting depth. Similarly, in Suter et al. (2021 [113]), mixing complementary species brought only benefits for forage production and quality, and N cycling under different rates of fertilizer. Therefore, the complex dynamic of vegetation groups working synergistically overlaps individual plant traits and improves the system productivity. 

The species richness and abundance of plant species have different responses to the ecosystem caused by their different functional traits [115,116], which affects the interactions between the system components and indicates the system organization [94]. In grasslands, biodiversity is driven by plant genotypes, changing not only in primary but also in secondary compounds, chemicals that attribute organoleptic characteristics of aroma, taste, and colour [117]. These compounds provide multiple benefits for humans, such as crop and livestock breeding, chemicals for medicines and materials for industry, and maintain biologically important ecosystem services, such as primary production, pollination, and nutrient cycling [7]. 

Plant secondary compounds may positively affect ruminant grazing animal and human health, with a vast range of benefits, including antioxidant and anti-inflammatory capacity [118,119], anthelmintics, antibacterial, antifungal, anticancer and antiviral properties [120,121,122,123,124]. Plant secondary compounds can also positively affect livestock animal products by the absorption and incorporation of these products into the animal gastrointestinal tract, thus providing some organoleptic compounds into milk or meat and enhancing their quality [31]. Diets with tannins for example, reduce the biosynthesis of skatole and its accumulation in meat and milk, which improve their flavour [125]. Modifications on rumen fermentation by consumption of those compounds also reduce negative environmental impacts from livestock production, such as CH_4_ and N excretion [126,127]. 

Other benefits of plant secondary compounds include enhanced pollination functional diversity, richness, and increased abundance of pollinators in the system [128], which contributes to ecological functions, such as plant phenological development [129].

The plant-herbivore interaction may also induce plants to develop tolerance traits that positively affect pollinators and optimise plant reproduction [130]. The biotic interactions between the soil microorganism community, plants, and ruminants and the system physiological responses to abiotic factors such as photosynthesis can be divergent between environments, thus having a differential influence on ecological processes [104]. Increased richness of plants in grasslands not only indicates higher levels of organization and vigour [131] but also acts as insurance for the function and stability of ecosystem services delivery due to the functional compensation among species in time and space [132]. Those benefits, besides increasing the resilience of systems, indicate the greater diversity of plants as an appropriate strategy for designing healthier grasslands. 

Phytodiversity is positively related to ecosystem function and plays an important role in grassland resistance to disturbances [133]. The more biodiverse the ecosystem, the more multi-functional and stable it is and the better its integrity [116,134]. This effect is expected due to a functional compensation among species in which their response to environmental fluctuations is asynchronous, which aside from assuring higher temporal stability in plants communities [115,135], ensures the resilience of the system. The resilience of a system is defined as the length of time that the system takes to recover from stress and its limit for absorbing various stresses [94]. Global climate change, extreme climatic events and climatic fluctuations, and livestock herbivory are examples of stress factors that singularly and together can threaten grassland health. In a study performed in a semi-arid savanna ecosystem in South Africa investigating response patterns in herbaceous abundances per functional group under different rainfall conditions and the presence or absence of herbivory, a shift in grass or forbs dominance was noticed across different situations [136]. The abundance of forbs dominated over grasses in the palatable annual functional group, possibly due to ruminant preference towards grasses, and forb species resistance against disturbance. For the perennial functional group, grass abundance was increased under above average rainfall conditions, but was co-dominated with forbs under conditions of average and below average rainfall and sustained long-term grazing pressure. Thus, forb species provided trait-based redundancy (functional equivalence in the ecosystem) by maintaining forage species stability during stress conditions when palatable grasses decreased in abundance [136]. Ecological responses to disturbances are highly variable and may have large effects on ecosystem functions [137] but rather than the extremity of the event, it is the resilience of an ecosystem that determines the intensity of ecological responses, which is dependent upon the traits of the species dominant in the ecosystem [138].

The influence of herbivory on plant dynamics is well known and can affect the vegetation community in different ways [139]. Large portions of above ground plant mass removed during grazing may decrease biomass production due to the decrease of leaf area and light interception, thus impairing photosynthesis [68]. On the other hand, ruminants grazing older living plant tissue or standing dead biomass may stimulate plant growth by exposing green and new leaves to the sunlight [140] and promote the return of organic matter to the soil as labile faecal material, thus resulting in greater nutrient availability for increased plant growth [141]. The mechanism that leads to plant compensatory growth following defoliation involves changes in plant physiology and modification of the environment around it [140]. A growth chamber experiment testing defoliation effects on Poa pratensis, a grazing tolerant grass, indicated that the increased C exudation due to defoliation quickly fed a growing microbial population that stimulated N availability and uptake and positively affected photosynthesis [84]. Under different grazing intensities in a desert steppe in Mongolia, a trend in vegetation composition changes was observed in grazing plant communities that have a long evolutionary history of grazing [66]. Authors reported that while the cover of some species decreased along the grazing intensity gradient, it increased for others. This was explained by the ability of species to compete for nutrients and survive in an arid environment, which also indicates their tolerance to grazing [66]. A grazing-induced change in native plant species dominance that did not coevolve with mammalian herbivores also occurred in the South Island of New Zealand, in which the growth and recovery post- grazing of *Chionochloa* spp. was retarded and allowed for more contemporary grazing-tolerant species to establish [110]. Therefore, as the magnitude of the impact of mammalian herbivores on plant communities is clearly variable in different environments with different evolutionary history (historical coevolution with different mammalian at different times), a better understanding of this area would facilitate the creation of design artefacts to enhance the knowledge of pasture carrying capacity to grazing intensity for designing more sustainable systems.

As grasslands are complex and dynamic systems, disturbances from herbivory are not always negative. A gentle and continuous disturbance is needed [142] to maintain vital ecological functions and grassland health [52], where ruminants can play an important role. Ruminants can be seen as one of the grazing strategies for achieving the meta-requirements of the design process as they contribute to enhanced productivity and biodiversity [143], nutrient recycling [142], elevate soil organic carbon and function, and benefits soil microbial species [52,55]. However, the positive impact of ruminant livestock on the ecosystem is dependent upon how plants are managed by humans and grazed by ruminants [144]. Native and non-managed swards or extensive grazing system are usually low productive and nutritionally deficient [3,145], and do not support as high levels of herbivory as productive systems. Thus, nearly all primary productivity returns to the soil as litter which explains the dominance of plants with ecophysiological traits best suited for this environment with low productivity and low quality [141]. Extensive and degraded systems also have a significant loss of soil organic C which decreases the function and resilience of soils over time [3,146]. In contrast, monotonic and introduced swards compromise grassland biodiversity and are highly dependent on external inputs such as fertilizer and land conversion [147]. This has been demonstrated in several experiments performed on European grasslands where multi-species sown pastures out-yielded monocultures of *Lolium perenne, Dactylis glomerata*, *Trifolium pratense*, and *Trifolium repens* at lower levels of fertilizer, indicating that the use of grass-legume mixtures can reduce fertilizer use and enhance the sustainability of grasslands [148,149,150]. Adequate management in pastoral-livestock systems should ensure sustainable land use in that the ruminant stocking rate and nutrient requirements meet forage biomass availability [151]. Ultimately, this highlights the importance of defining grazing management to control herbivory disturbance based on the soil and vegetation features of the landscape to maintain the ecosystem integrity and enhance its resilience. 

Grassland management affects animal grazing decisions, and both can influence ecosystem function. Grazing decisions aim to meet animal nutritional requirements, which are dependent upon nutrient interactions and the physiology of the animal [152]. Ruminants have individual food preferences [153] but their diet is selected according to plant properties, quality and abundance offered in the landscape [154,155]. In natural grazing and browsing habitats, grasslands are diverse with different species and arrangements varying in primary and secondary compounds [156,157]. Thus, ruminants tend to select mixed diets with an enhanced variety of nutrient composition. That variety also extends to phytochemical compounds that promote benefits to their health [158]. In pastoral-livestock systems, ruminants will most likely act as in natural habitats when searching for diets that meet their own individual requirements. However, if the ‘herbage’ does not match an individual needs due to a lack of diversity or quality, nutrients and medicine supply, then performance is compromised [159,160], with further consequences to the environment, such as overgrazing and excessive stock camping due to the unequal grazing distribution on the landscape and ruminants exercising selective grazing [161,162]. Grazing behaviour needs to be considered in grazing management by providing ruminants with a diverse and nutraceutical diet to not only satisfy animal requirements but reduce environmental negative impacts nutrition-related and therefore, contribute to preserving and enhancing grassland health. 

Well-managed pastoral-livestock systems contribute to the stability (organization) of the ecosystem, resilience, and productivity (vigour) by providing more essential ecosystem services [151]. Once herbivores and grasslands coevolved [163] in most grassland ecosystems, concepts based on animal behaviour and ecology domains (kernel theories) help us to explain the complexity behind the interaction between ruminants and grassland towards making better grassland management decisions. The natural movement of large herds across landscapes characterised by short periods of heavy and uniform use of grass species and followed by ruminants absence allows plants to regrow between grazing times [3]. The time between grazing is called the recovery period, it allows plants to complete restoration of root reserves and accumulate nutrients [164]. Grazing management should provide plants with a period of vigorous regrowth before the next grazing [151] and therefore, enable high nutritional value and productive pasture [165] that improve animal performance, increase soil organic matter and C accumulation, and reduce environmental impacts [166,167]. Plant science is one of the kernel theories needed in the design framework for a better understanding of plants physiology, growth development and adaptive response to ruminants towards enhancing the health of grasslands under livestock pastoral production systems. Knowing the cultural-historical evolutive processes of a landscape and how it defines current soil and vegetation features, facilitates the best decision on how to manage grasslands considering the plants-ruminants relationship and its influence on grasslands health.

### 2.3. Ruminants

Ruminants can play an important role in the ecological processes that promote ecosystem services and enhance grasslands health. Ruminants provide the soil with nutrients not only from urine and faeces but also from defoliation. The input and decomposition of organic matter from litter and root exudates on soil increase the availability of nutrients for microbes and plants, with a higher allocation of C and N mineralization, thus promoting a faster plant growth, with increased aboveground net primary production and improved plant quality [141,168,169,170]. Besides being beneficial for grazers, increased primary productivity and plant quality contributes to ecological processes and ecosystem services. The increased C input to the soil during defoliation stimulates the microbial activity and the formation of microbial-derived persistent carbon [146,171], which contributes to long-term C sequestration on ecosystems [172]. By grazing, ruminants also reduce the flammability of aboveground biomass, redirecting the combustible biomass to a source of belowground organic matter, which besides reducing fire events, promotes the shift of C from more vulnerable storage to more persistent ones, increasing C persistence on grasslands [103,172]. Nevertheless, those benefits are context-dependent and are only achieved under certain vegetation, climatic, and soil conditions and their interactions, besides the evolutionary history of grazing at the place [103,141,168]. Defoliation may also cause a change in plant functional communities, which may be positive and enhance primary production through plant compensatory growth; or negative, when, in unfertile and low productive soils, the shift occurs from palatable plants to species that will better fight against herbivory and produce a poorer quality litter, consequently, reducing soil microbial biomass and activity, nutrients availability and plant productivity [170,173]. Therefore, to better understand the influence of herbivory on grasslands because of the relationship between ruminants, plants, and soil, this section will review this interaction from a ruminant point of view. 

Grazing activity is a complex process nested in time and space [174] in which ruminants modulate their behaviour with the aim of searching for nutrients that will be converted into chemical compounds in the rumen as their main source of energy [175]. Intrinsic and extrinsic factors trigger an individual motivation to graze, from which outcomes affect their performance, health, and the environment. When grazing, ruminants attempt to satisfy their individual food preferences and demands whilst avoiding toxic and unliked plants [176]. Their final choice is based upon the palatable characteristics and biochemical composition of swards and on the post-ingestive effects animals have from previous experiences [177]. For instance, it is well documented that ruminants exhibit a preference for legumes and have a higher intake rate compared to grasses [178]. Accordingly, they spend more time grazing when the pasture is composed of legumes and grasses rather than only grasses [179]. Still, ruminants recognize the post-ingestive effects of eating certain foods and adjust their intake at safe levels [180]. As an example, a study showed that when ruminants had the option of eating a pure white clover diet, they chose to include some grass in their diet to avoid bloating or other toxic consequences from legumes [178]. These factors mean ruminants are highly selective animals and are committed to search for forages that meet their requirements, which is why grazing decisions, which include behaviour, physiological, and strategy changes [175], are an expression of ruminant response to an environment. As grazing decisions are affected by the imposed grazing management ([165], the dominant grazing regime can therefore indicate the condition of foodscapes and landscapes [155], thus indirectly contributing to the assessment of grassland health in pastoral-livestock systems, in addition to providing insights about ruminant performance and health. Therefore, ruminant grazing behaviour is a key tool and an essential kernel theory to understand the relationship between plants and ruminants and to modulate management towards enhancing grassland health. 

When the herbage that ruminants are grazing fails to provide individual animal nutrient requirements, the animal performance [181], health [182,183], and welfare [184,185] may be compromised. Animal nutrient demand changes according to their productive life stage and health status and varies from individual to individual. When trying to acquire the required nutrients from non-balanced pasture, ruminants will most likely face two scenarios. They will either stop eating when they have satisfied some of the nutrients, while failing to achieve others, a phenomenon known as incidental restriction, or they will continue eating to satisfy all the nutrient requirements, what may lead to overconsumption of some nutrients, a phenomenon known as incidental augmentation [154,186]. An excess of magnesium, potassium, and manganese, and a deficiency of phosphorus, sodium, copper, and iron was reported in a study simulating nutrients intake of cattle from pasture [187]. Considering the importance of trace elements and vitamins for health and productivity, nutrient deficiency, or toxicity, depicted by different symptoms, affects ruminant physiology and reproductive performance [181]. Thus, an adequate balance of food avoids malnutrition and further negative effects [182,183]. Malnutrition is also intimately related to oxidative stress [153] which makes the animal more susceptible to other diseases [188,189]. Both herbage offers and choice consequences can impact animal welfare. Ruminants face the challenge of meeting their nutritional demands and building a balanced diet while grazing [185]. This task may lead to either positive or negative consequences for animal behaviour and welfare. If animal demands are satisfied, positive emotions are generated, as ruminants were given the challenge, but also the opportunity of efficiently behaving and solving problems, which is positive to their welfare [185]. On the other hand, negative experiences may result from an unsuccessful search, with detrimental outcomes for animal behaviour patterns, welfare, and performance. When ruminants are unable to achieve their individual demand, they can get frustrated and stop eating, or even develop an aversion to foods that are too frequent in their diets, with further reduction of intake and consequences for productivity [182,184].

As ruminant nutrition is intimately related to the environment they graze from, nutritional strategies are paramount for designing healthier grasslands. A disequilibrium in the rumen from inadequate nutrition and inadequate energy and nutrient use alters rumen fermentation and products [190], which may increase the production and emissions of enteric CH_4_ gas [191]. On the other hand, a high nutritional value diet leads to a reduction of enteric CH_4_. In pasture, this is reached during the end of the ascendant phase of the sigmoidal growth curve of plants in which plants have the highest protein and lowest fibre content, as well as the highest productivity rate, thus increasing animal performance and mitigating CH_4_ emissions [89,165]. Nevertheless, different forages have different chemical composition—legumes versus grasses, C_4_ grasses versus C_3_ plants—and different secondary compounds, therefore, are divergent regarding their effects on CH_4_ production [159,192,193,194]. Likewise, N excretion as urine and faeces is an environmental issue for pastoral livestock production systems. The excess N ingested, because of an imbalanced diet, is wasted and released through urine and faeces, which can cause air and water pollution [195]. Furthermore, as previously mentioned in other sections, the grazing activity of ruminant livestock can degrade pasture and soils. As ruminants are highly selective, they may spend more or less time grazing as a compensatory strategy in order to select their diets and meet their nutritional requirements due to the variable pasture structure in given grazing areas [165]. Consequently, some areas may be overgrazed while others are avoided. Overgrazed areas degrade plants and soil [73], and under grazed areas allow non-desired plants to grow, thus changing the dynamic of plant communities [141], with further consequences on swards nutritional value and biomass production, and so, on grasslands health.

The quality of ruminant products (e.g., milk and meat) is influenced by the taxonomical and biochemical composition of the swards ruminants are grazing [31,196]. This is due to the secondary compounds of plants such as phenolics, terpenes, and fatty acids that enhance the flavour and biochemical characteristics of meat and milk and can offer medicinal properties to animals and humans [158], such as anticarcinogenic, anti-inflammatory, antioxidant, anti-diabetic, anti-obesity and anti-atherogenic properties [197,198,199,200,201,202]. Those benefits are enhanced in highly nutritive and diverse pastures [31,89,196]. When comparing a semi-natural pasture composed of purple moor-grass *Molinia caerulea* with improved pasture composed of ryegrass *Lolium perenne* and clover *Trifolium repens*, meat quality was enhanced with lower proportions of saturated fatty acids and higher proportions of essential polyunsaturated fatty acids such as C18:2n−6 and C18:2n−3 [203]. The concentration of these compounds of interest also change according to the pasture growth stage [204]. Kuhnen et al. (2021 [205]) in comparing three different cutting intervals (38, 54, and less than 31 days) found that the latter interval (54 days) resulted in lower levels of carotenoids, flavonoids, and phenolics, which would probably result in lower levels of those compounds in animal products. Therefore, as the provision of nutritious food is also an ecosystem service from grasslands [5,6,7], the impact of animal nutrition on product quality is also a factor to be considered when designing pastoral livestock production systems to enhance grassland health.

Considering the effects that pasture has on ruminants and vice versa, grazing management should account for not only adequate nutrition offered to ruminants but also for ruminant grazing decisions and behaviour. Ruminant nutrient requirements and feeding patterns change in the contexts of space and time [177]. Likewise, nutrients in plants are highly variable across species and spatial and temporal contexts, and single species pasture are unlikely to attend to all of an animal nutrient demand [156]. Grazing management should thus be done accordingly. Offering a diverse pasture is more promising in satisfying ruminant demand and preferences, besides stimulates their intake and improves their performance [29,184,206]. Ruminants would be able to select their diet and benefit from a taxonomic and biochemical variety of plants with different nutraceutical values [207], including the selection of plants rich in secondary compounds with medicinal properties to self-medicate [153,208]. They would also be able to identify and avoid antinutritional and toxic components of plants [157]. With a higher chance of ingesting adequate proportions of nutrients and building a balanced diet, N excretion [29] and CH_4_ emissions [126] would potentially reduce. A diverse selection of available forage species shows more potential in promoting the expression of explorative and selective behaviours in ruminants [185] and providing them with satisfaction and comfort [153], while improving their welfare.

After exploring the role of the key components of grasslands and how their interaction influences grassland health, the meta-requirements, the meta-design, and the kernel theories of the design framework can be defined (Figure 2).

## 3. Design Method

We previously described how the key components of grasslands (soil, plant, ruminant) contribute to the construction of the design framework by providing support to define the meta-requirements, the meta-design, and the kernel theories. Based on that, the design method section will outline what features of soil, plant, and ruminant support the construction of the artefacts for the design process.

Pastoral livestock production systems can be managed to positively affect and mediate ruminant health and performance while decreasing environmental impacts andcontributing to enhance grassland health. Understanding how grassland components perform to offer ecosystem services, allows for the definition and description of the artefacts required to achieve the design goal. Under pastoral livestock production systems, the artefact to enhance grasslands health is a strategic and climate-topographically contextualized grazing management that considers the interaction between soil, plants, and ruminants (Figure 2). Overall, strategic grazing management would provide grasslands with abundant and phytochemically diverse swards that would promote nutrient cycling for vigorous and high-nutritive pasture for assuring animal performance, welfare, and health, with consequently high-quality animal products, while reducing environmental impacts. Plant species diversity would benefit the closely integrated community of soil, as well as the system biodiversity and resilience, as previously stated in this review. An adequate distribution and pressure of ruminants grazing across the whole area would reduce soil erosion, and other impacts such as water contamination. The species of plants would be complementary and well suited or adapted to the climate conditions. Seasonal and topographic features would be considered to define the animal rotation.

As there is a diverse set of grazing systems existent, the best strategic grazing management is the one tailored to the context [209]. A meta-analysis comparing different grazing systems and the impact on vegetation, soil and livestock productivity parameters indicated that the outcomes are divergent worldwide and dependent on several factors [42]. In general, results suggested that vegetation was mostly negatively affected in continuous grazing, but this negative effect was less under light or moderate grazing levels and high precipitation conditions; while a higher probability of negative effects on soil properties were noticed in long duration studies of rotational grazing. In addition, multi-paddock systems resulted in fewer negative impacts on vegetation and livestock productivity, whereas the opposite was observed for short duration grazing systems. Controversially, a study performed in Colombia under short duration grazing systems indicated higher pasture and animal productivity with further positive impacts on soil quality [41]. In another meta-analysis comparing the performance of holistic planned grazing versus seasonally continuous grazing at moderate stocking rates in different countries and ecosystems, no difference was found regarding plant and animal productivity [210], which can be a result of erroneous selection of grazing factors such as low stocking rates, long occupation periods and/or inadequate recovery period for the pasture. In the Piedmont region of Georgia, United States however, a continuous grazing system approach was more detrimental for soil health and increased runoff compared to strategic grazing [211]. The authors defined and described the strategic grazing as a mix of multiple management practices that considers the rotation of ruminants into smaller paddocks to allow forage growth, manure distribution, and exclusion of vulnerable area of pasture, followed by overseeding with productive forage mix. On the other hand, Drewry (2006 [212]) in a review about natural recovery of pastoral New Zealand and Australia soils concluded that the complete exclusion of ruminants for months or years would be recommended for reducing soil damage during wet conditions.

These contrasting results indicate that there is not a necessarily/fundamentally better or right system, but each system has benefits under the right conditions [209]. This implies that regardless of the grazing system per se, grazing management tools must be carefully chosen and adjusted under a set of specific conditions as set by soil, climate, topography, and plant communities. Grazing management tools include the choice of plants species best suited for the soil when sowing is required, residual herbage mass and rest length, seasonality, stock type more adjusted for the landscape, adequate number of ruminants grazing per area (stocking rate) to guarantee an homogeneous distribution over the landscape, and timing of grazing (occupation time). Although, those tools follow a pattern for each grazing system, they are variable according to the landscape and farm management. To illustrate, a comparison study between three different recovery periods (short, medium, and long) for mixed swards in a rotational grazing situation in Brazil during Summer, resulted in more productive and higher nutritional value herbage with reduced in vitro CH_4_ relative emissions for the shorter period [165]. Fast rotation (56 days) was also more productive in terms of animal productivity in a rotational grazing system in Australia, during the same season (Summer) as the previous work [213]. Although both studies indicated short paddock rest periods, the rest length in days varied, with 24 days for the former and 56 days for the latter. This demonstrates that the similar system and grazing tools during the same season in countries with similar climate conditions may still present differences. Those differences may be attributed to one of the follow factors: soil type and properties, plant species, cattle breed, and/or their interactions.

The consequences of using the same/similar grazing tools and management in different contexts may be difference in magnitude. This was observed in a study that compared the effect of different beef cattle grazing intensities on sward condition in four different grassland types in Hungary [214]. In that study, Török et al. (2018 [214]) reported a significant interaction between grazing intensity and grassland type on vegetation parameters manifested at different magnitudes, confirming that the context is a strong determinant of the performance of grazing systems. Herrero-Jáuregui and Oesterheld (2018 [215]) conducted a similar study with different stock types (cattle, sheep, goats) and determined that most negatively affected grasslands were in arid, low productivity systems. These aforementioned examples demonstrate that significant impacts of similar grazing intensity were observed in different environments, but these impacts affected grassland health at different magnitudes. Therefore, grazing management should be planned strategically according to the local conditions. Having previous studies and experiments as models to establish grazing management in other places are essential in the decision-making process, where the best tools would be replicated in order to achieve similar or better outcomes. However, as climatic-social, and topographical conditions of grasslands are highly variable, the choice of best grazing tools, regardless of the grazing system, remains a challenge. The testable design hypothesis from the design framework would contribute to the evaluation of tools and estimation of results before putting them in action, and therefore, facilitate the final choice.

## 4. Testable Design Process Hypothesis

The testable design process hypothesis is the method used to verify whether the meta-design satisfies the meta-requirements, and whether the meta-design and design method are consistent. Most methods used in natural systems such as pastoral livestock production systems are field experiments that vary in geographic-climate conditions, duration, grazing systems, and management tools. The purpose of field experiments is to reproduce the natural conditions imposed on ruminants in order to test a hypothesis, nevertheless, due to the complex and dynamic nature of grazing systems, there are many factors that need to be accounted for to reduce the influence of experimental error, where possible. For example, resource heterogeneity can bring different outcomes to ruminant population in different environmental contexts and provide erroneous or limited assumptions about the herbivore-vegetation relationship in general [216,217]. The experimental design needs to incorporate statistical criteria of the study that guarantee the visualization of any resulting patterns and trends to thus allow accurate interpretation. Furthermore, the independent variables in field experiments need to be carefully considered, as they are random factors difficult to totally control and may influence the results. Researching individual components of the system may also lead to reductionist approaches that generate sketchy results, as the reductionist approach contradicts the complex nature of these systems [32]. Thus, holistic and multidisciplinary approaches could be more suitable [218]. Other attributes that need to be addressed for conducting scientific experiments include a clear definition of the problem, objective, and hypothesis; the choice of the most appropriate ruminant models; number of ruminants; time frame; familiarity with the techniques and subject [219].

In grazing and foraging ecology and management of grasslands, another strategy to acquire the answers to hypotheses is through meta-analysis, which are statistical reviews that combine different studies around pre-defined problem statements. In a review about the use of meta-analysis in Animal Science, Sauvant et al. (2020 [220]) argues that this strategy is proven to be efficient and largely accepted, although it also requires several considerations (such as the publications selected to be analysed), for drawing accurate conclusions. Results from meta-analysis can also be used in experimental models to estimate further results, or develop new models, which reinforces the need to use accurate inputs.

Modelling is also an efficient and widely accepted way of verifying hypotheses within pastoral livestock production systems [221]. Through simulation modelling, more context/site variables can be addressed and accounted for using spatial and time variability assessment of the desired variables, without the time and financial cost restrictions found in field work [52]. Modelling also allows for the wider exploration of different situations not always viable with field work such as different grazing tools—recovery period and stocking rates [222]. Moreover, the use of modelling promotes the simulation of grazing management strategies and anticipation of outcomes before putting actions into practice.

Another strategy is the use of Geographic Information Systems (GIS), a computational system that allows the analysis, creation, and management of spatial data. Geographic information systems are widely applied to Animal Science questions and can be used for land use/cover changes [223], land degradation [224] and grassland health assessment [4,8] by spatial analysis of multi-temporal data from a particular area. Geographic information systems generates a diverse range of information that can be integrated to produce useful outcomes that can contribute to the decision-making process [225] and allows the identification of suitable areas for specific purposes, such as adequate areas for specific plant species growth [226]. The use of GIS offers an evaluation and diagnosis tool for grasslands accounting for spatial and temporal variability; thus, providing insights about landscapes, and contributing to management decisions.

All the aforementioned strategies applied to Animal Science parameters assessment can be used in the testable design process hypothesis. It is important to choose the most adequate strategy for a particular context and be aware of the pros and cons, considering all the appropriate variables that may affect the results.

## 5. Design Framework and Further Applications

This literature review has contributed to a more holistic perspective of grasslands and a better understanding of grasslands components and processes. It has done this by considering systems thinking theory principles, for the creation of a design framework to diagnose and enhance grassland health under pastoral livestock production systems. A general framework was created (Figure 3), based on design theory from Walls et al. (1992 [28]) that would be a guide to determining the artefacts necessary to achieve specific goals to enhance grassland health, the construction of those artefacts and the required theories behind them. In addition, the framework provides a method to verify the validity of the design process. It is important to clarify that the framework does not provide specific tools, such as plant species or stock density, as those are context specific. Grasslands are complex environments from which health is a consequence of how the components interact. In general, the structure of all grasslands is similar—same components, ecological processes, and ecosystem services, but with different specifications (e.g., different plant species, different soil types, different climatic conditions), that interact with different environments. Although similar patterns may be noticed, minimal variations between particularities within the system and its natural dynamism, create a complex heterogeneity that results in completely divergent ecosystems. That is the reason why creating very specific design frameworks would be limiting and could provoke sketchy interpretations and application.

## 6. Conclusions

The goal of this literature review was not to develop ready solutions for regionally or geographically specific pastoral livestock production systems but rather promote a facilitating tool to enhance the understanding of grasslands complexity among stakeholders and help with the creation of strategies that account for the holistic nature of grassland systems. A general framework arises more as an instructed exercise where users are provided with a guide to better understand how the complexity within grasslands occurs and the implications of it, without offering specific and direct tools. Pastoral livestock production systems need to be context adjusted, considering soil and plant characteristics in response to local abiotic factors and grazing ruminant response and impacts. Accordingly, the framework allows for the flexibility of being used in different environments and being extended for any design purpose of enhancing grassland health under a pastoral livestock production system. Therefore, designers may use the framework as a facilitating tool, following this structure for the planning of actions and grazing management decisions to design systems that evolve continually with healthy grasslands to sustain human life.

As an example of applying the design framework is the case study of a pastoral livestock production system located in the South Island rangeland of New Zealand. The heterogeneous hill and high-country landscape of Mount Grand station is characterised by highly variable topography in terms of land aspect, elevation, and slope, with significant variation in the soil condition and vegetation community. Consequently, the functions of that ecosystem, such as primary productivity, herbage feeding value, and plant secondary compounds are also highly variable, with further divergence in the interaction of soil-plants-ruminants. Therefore, the current grazing regime(s) on the station would need to be adjusted for these discrepancies, as the same grazing strategy would result in different outcomes with potentially unknown consequences for the health of the grassland.

The first step in the design process of this hill and high-country station would be the understanding of all the system components and their interactions with the system behaviour, the creation of a rich picture as systems thinking approach would be a good starting point. The rich picture can be created by using GIS and field work embedded in a multidisciplinary context yielding a graphic representation of a mental model towards the diagnosis of the current health condition of the station (See Pereira et al., 2021 [8]). The rich picture would then lead to the process of investigating, thinking, and defining the construction of the grazing and browsing management as an artefact of the design process for achieving the meta-requirements for enhancing the grassland health of the station. In this case, the creation of a rich picture enabled the simultaneous visualisation of different information (e.g., plants species, abundance, soil chemical, physical and biologic properties, land aspects, slopes, elevation), promoting a multidisciplinary discussion and a better understanding of the impact that the interactions of these components have on the condition of the station.

The artefact noted above is the grazing and browsing management strategy, but its construction depends on the grazing strategies and tools chosen that best suit the context of the station based on outcomes from the rich picture. These strategies (plant species, grazing level intensity, stock type, paddock allocation) characterize the design method and need to be defined according to the applied kernel theories in the base of ecology, soil and plant science, and ruminant nutrition and behaviour. As an example of grazing strategies and tools, different stock types at different stock densities and grazing days on paddocks along the station across the year would characterise different grazing and browsing management with divergent results. Likewise, rotating ruminants within the station based on the characteristics of the vegetation community of each paddock could be another grazing strategy. All the strategies would need to be defined and tested to guarantee that they positively respond to the meta-design, towards the achievement of the meta-requirements before putting them into action. Therefore, simulation modelling exercises, and proof of concept short trials (e.g., nutraceutical effect of on native phytochemical rich plant/communities of plants sequence of consumption on animal health) should be performed to support the simulation of different scenarios during the design process.

The outcomes of the testable design process hypothesis (simulation modelling, proof of concept) would show whether the design method (grazing strategies and tools) and the defined artefact (grazing and browsing system) are aligned and adjusted for the context of the station, towards the design goal (enhance grassland health), and thus contribute to the decision-making process.

## Figures and Tables

**Figure 1 animals-12-03306-f001:**
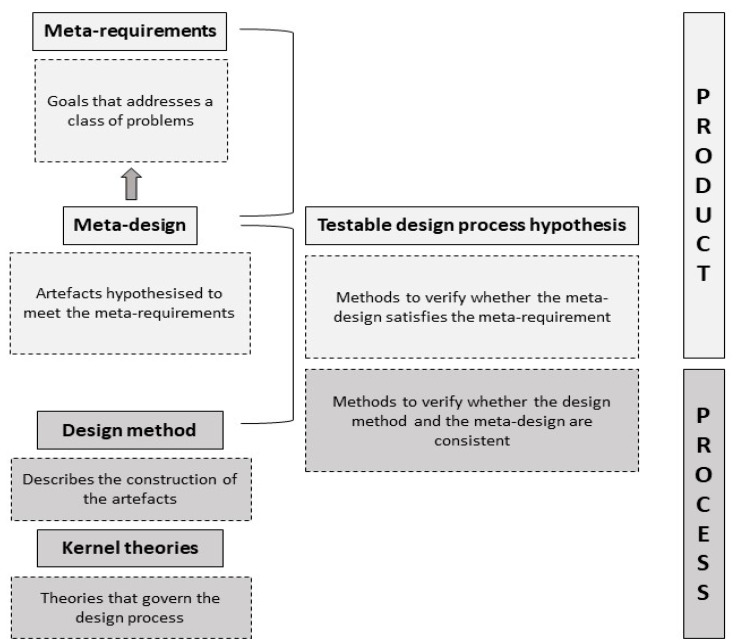
The framework of the design product and process of the design theory proposed by Walls et al. (1992 [28]).

**Figure 2 animals-12-03306-f002:**
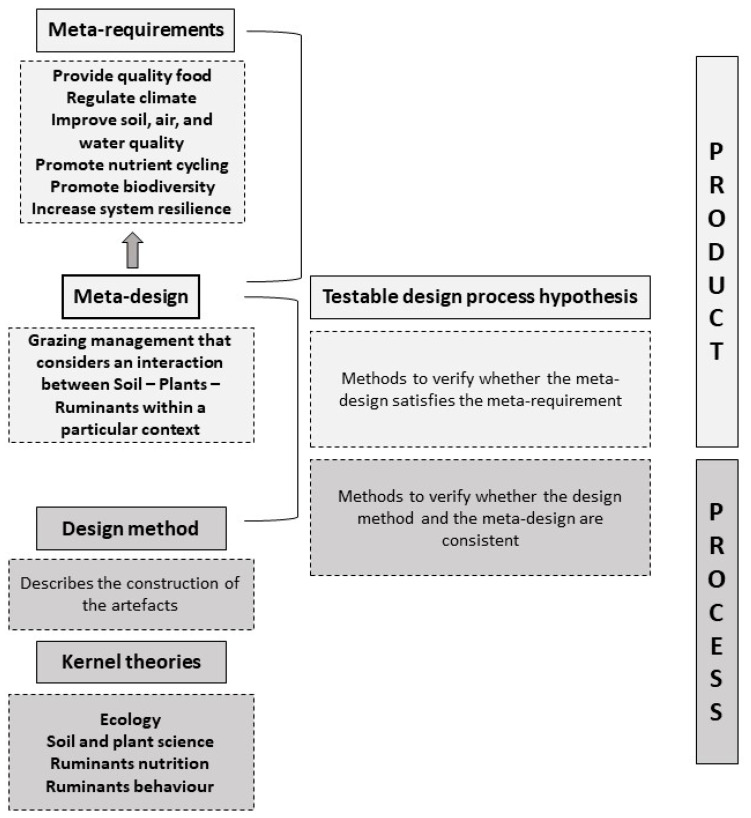
The framework of the design product and process with a description of the meta-requirements, meta-design, and kernel theories for designing healthier pastoral livestock production systems. Based on the design theory proposed by Walls et al. (1992 [28]).

**Figure 3 animals-12-03306-f003:**
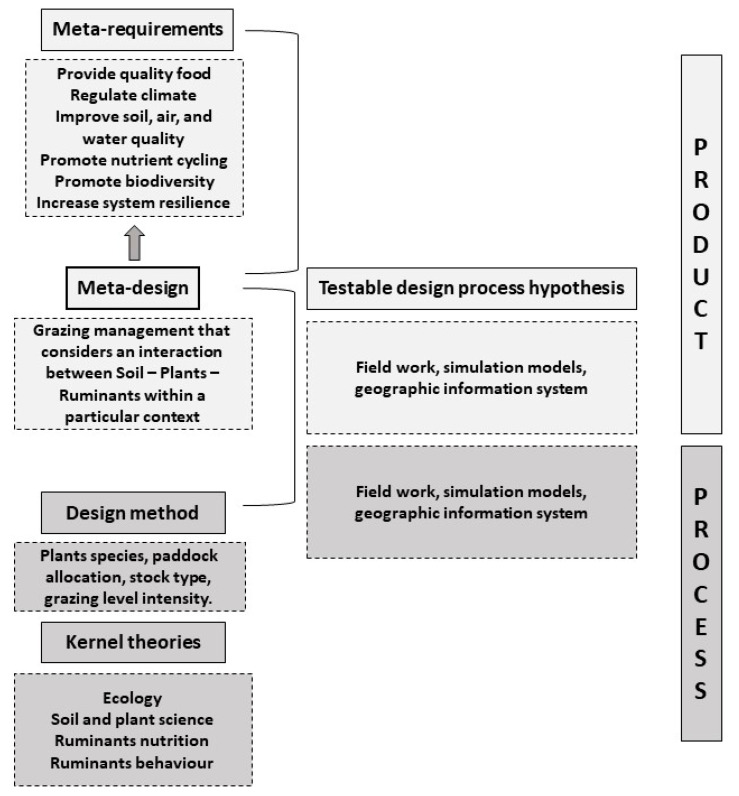
The design framework to diagnose and enhance grassland health under pastoral livestock production systems based on the design theory proposed by Walls et al., 1992 [28].

## Data Availability

Not applicable.

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
