# Peer review of "Creating a Design Framework to Diagnose and Enhance Grassland Health under Pastoral Livestock Production Systems"

_animals, 2022, doi:10.3390/ani12233306_

Round 1

Reviewer 1 Report

I appreciated particularly a review with the intent to create a framework to design alternative and more sustainable pastoral livestock production systems.

The research approach of this paper is innovative. This systems thinking method in my opinion represents a new interesting way through an holistic theory. It passes through a narrative review, for the creation of more sustainable livestock production systems with an overview of grasslands and the final objective of creating pastoral livestock production systems that enhance grassland health.

I think the conclusions consistent. The outcomes of the design process hypothesis are well presented for the final design objective, i.e. the enhancing of the grassland health generally contributing to the decision-making process.

Line 244: what is Al? …but not for Al, which was higher in the upper slope soils…
Line 664: please explain in a short sentence…which brings positive welfare outcomes [185].
Line 697-698: please consider other citations: e.g., Renna M., Ferlay A., Lussiana C., Bany D., Graulet B., Wyss U., Enri R.S., Battaglini L.M., Coppa M. Relative hierarchy of farming practices affecting the fatty acid composition of permanent grasslands and of the derived bulk milk. Animal Feed Science and Technology, 267, 2020, 114561.
Line 810-811 Check the sentence
Line 967-968 Maybe useful to cite Renna et al., Animal Feed Science and Technology, 267, 2020, 114561
Line 1305 : the year must be in bold

Author Response

Thank you for your comments and suggestions. We have uploaded a document with point-by-point responses.

Reviewer 2 Report

[Animals] Manuscript ID: animals-1996553 - Creating a design framework to diagnose and enhance grassland health under pastoral livestock production systems

General Comments

This is an extremely long paper, and is probably too long. I found that there is a lot of repetition in some areas. I would suggest that the authors try to shorten the paper. I have highlighted some areas where there is repetition but not all. I would urge to authors to go through the manuscript to try to shorten the paper if possible.

The paper is generally well written but there are a lot of long sentences and sentences that are phrased poorly. Again I have highlighted some but not all of these instances. Please go through the paper and review.

There is a lot of interchanging of the terms animals ruminants and herbivores. Is it possible to go through the paper and perhaps be more consistent with the use of one term? Is ruminants the correct term for the heading under the systems components? Perhaps animals might be better?

Specific comments

L89-91: Rephrase this sentence, particularly the latter half of the sentence please? The meaning of this section of the sentence is not clear to me “which implies the necessity of farming systems in embedding agricultural activities in a social-ecological-geographic perspective [26].”

L100-103: Rephrase this sentence please as it is worded poorly. Suggest “For example, pasture swards can be designed to provide animals with a diverse forage that is biochemically rich, which can enhance ruminant systems production and environmental performance [29, 30].”

L202-204: Rephrase this sentence please.

L217: “…C input, ground cover and the soil micro and macro…”

L233: “…define a soils ability…”

L236-238: Rephrase this sentence please.

L300: Replace “Whereby” with “Where” here please

L326-327: “…, which liberates limiting nutrients, such as N, for plant regrowth [84, 85] and also improves soil structural stability and resilience [76].”

L405-409: Split this sentence into 2. “In the Mediterranean region, trees, shrubs, and forbs were found along the whole climate range, but their abundance and pattern traits, such as leaf size, texture, thickness, and type, changed according to climatic gradients. Plants with smaller leaves are more abundant in cold and dry environments, whereas coriaceous plants with a thick leaf texture are more abundant in more aridity environments [109].”

L410: Replace “roots” with “root”

L412: Replace “with” with “which”

L426: “Functional diversity increases system…”

L446: Reword this phrase please “genetic material variety of plants”

L465: “the plant-herbivore…”

L514-518: This sentence is poorly written, the authors move between tenses within the sentence and it is too long. Split this sentence in 2. “Authors reported that the cover of some species decreased, while others increased along the grazing intensity gradient. This was explained by the ability of species to compete for nutrients and survive in an arid environment, which also indicates their tolerance to grazing [66].”

L521-525: Very long sentence with some confusing phrases in it. Please replace “mammalian herbivore” with “mammalian herbivores” and clarify what “different historical land” and “knowledge of plant carrying capacity to grazing intensity” mean please.

L600-610. I question the usefulness of this section here. It is repetition of the C storage/sequestration points made previously and should this be in the ruminant section.

L629-630: “Accordingly, they spend more time grazing when the pasture is composed of legumes and grasses rather than only grasses [179].”

L630-632: I don’t agree with this statement and I’m not sure that is exactly what the references you refer to state. Chapman et al. (2007) refer to a 70:30 preference for clover compared to grass and state that the intake of clover may be limited by some physiological factors but they do not state that animals adjust their intake at safe levels. Please revise this statement.

L653-655: Again I think this is a stretch based on the reference you have cited. I’m not sure that such high levels of N intake to cause ammonia toxicity and death can be achieved from a forage diet so I don’t think that this statement is useful here.

L671-672: “A disequilibrium in the rumen…”

L700-702: “Highly nutritious and diverse species pastures have been shown to be richer in bioactive compounds with positive benefits for those products and hence, human health [89]”. Please clarify what are the positive benefits you refer to and clarify what you mean by the term “products” do you mean the bioactive compounds?

L708: Please clarify what you mean by “latest cutting interval”

L720-726: Rephrase these sentences please especially the phrase “Self-medicate when in illness condition”

L750-751: “while decreasing environmental impacts and contributing to enhance grassland health.”

L772”high precipitation conditions”

L781: Do you mean “which can be a result of erroneous…” rather than “what can be…”

L800-801: Please clarify what you mean by “as well as the way of how 800 animals are distributed over the landscape.”

L815-821: Rephrase this sentence and possibly split into 2 sentences

L903: Delete “and delivery ecosystem services”

Author Response

(The authors gave the same response as above.)

Round 2

Reviewer 2 Report

Thank you for revising the paper in line with my comments. Please have a final proof read for some minor grammatical errors and typos. Thank you.